# Enhancing Operating Efficiency in China's High-End Equipment Manufacturing Industry: Insights from Listed Enterprises

**Yi Zheng * and Min Luo ***

School of Economics and Management, Shanghai Ocean University, Shanghai 201306, China
* Correspondence: yzheng@shou.edu.cn (Y.Z.); m200501193@st.shou.edu.cn (M.L.)

**Abstract:** The high-end equipment manufacturing industry is a strategic sector for China's manufacturing transformation and upgrading. However, this industry is facing a series of challenges, such as insufficient innovation capabilities and poor business operations. This paper uses the super-efficiency SBM model to calculate the operating efficiency of listed companies in this industry from a micro perspective and conducts in-depth multi-angle analysis of their operating efficiency. Furthermore, Tobit regression is utilized to identify the factors that affect operating efficiency. The aim is to provide a pathway for companies in this industry to achieve efficiency maximization and sustainable development. The research shows that the average operating efficiency of high-end equipment manufacturing companies was around 0.7 from 2016 to 2021, and nearly 70% of companies were in a non-DEA efficient state. The operating efficiency of the intelligent manufacturing equipment industry is far higher than other industries, and the western region has great development potential. In addition to government subsidies, factors such as company age, equity concentration, regional GDP, and regional openness all have a positive impact on the operational efficiency of high-end equipment manufacturing companies. This paper combines the characteristics of the equipment manufacturing industry and analyzes their operating efficiency from multiple dimensions, providing decision support and pathways for the high-quality and efficient development of this industry.

**Keywords:** high-end equipment manufacturing; operating efficiency; super-efficiency SBM; government subsidies

## 1. Introduction

The prevailing trend in the current industrial revolution is to advocate for Industry 4.0, where countries strive to augment production efficiency across multiple industries through the development of cutting-edge technologies, renewal of equipment, and digital transformation [1,2]. The United States, Germany, Japan, and other developed countries have established themselves as leading powers in high-end equipment manufacturing industry technology, products and services [1]. China's high-end equipment manufacturing industry has a relatively late start, with underdeveloped basic technological conditions and a heavy dependence on imported key components [3,4]. Its independent innovation capacity is weak [5], and there is a high level of foreign dependence on key core technologies [6]. The level of internationalization within the industry is insufficient, with enterprises having limited ability to operate on a global scale and waste of resources [7]. Additionally, challenges such as an imperfect manufacturing innovation system and inadequate product safety measures persist. The industry is caught between the advanced level of developed countries and the rapid development of developing countries, where it faces immense development pressure and formidable obstacles to industrial upgrading [8]. The Fourteenth Five-Year Plan and Vision 2035 Outline identifies the development of high-end equipment as one of the key industries in achieving China's goal of scientific and technological power and proposes to increase investment in research and development, improve the quality and efficiency, and strengthen the integration of information technology and manufacturing,

among other strategies [9]. High-end equipment manufacturing enterprises, under the guidance of various policies, are not only supported by the local government but also receive various types of government subsidies [10]. Achieving optimal resource allocation using limited resources and adjusting the scale to maximize output, efficiency, and sustainable development of enterprises has become a pressing concern for China's high-end equipment manufacturing industry.

The aim of this paper is twofold. Firstly, it seeks to investigate the state of operating efficiency among high-end equipment manufacturing enterprises in China and explore potential differences between various industries and regions. Secondly, it aims to examine the factors that impact the operating efficiency of high-end equipment manufacturing enterprises and whether government subsidies, as a policy measure, are beneficial to improving the operational efficiency of high-end equipment manufacturing enterprises.

Possible contributions and innovations of this paper include: (1) inclusion of R&D investment as an input variable and the number of patents applied as an output variable in measuring the operating efficiency of high-end equipment manufacturing enterprises, taking into account the knowledge-intensive nature of the industry and the importance of innovation capability for future development. Incorporating a company's technological innovation into the calculation of its operational efficiency can better measure the development potential of the high-end equipment manufacturing industry. (2) Examination of the impact of government subsidies on the operating efficiency of the high-end equipment manufacturing industry, exploring whether there are differences across industries and regions, and providing guidance for local governments in developing policy measures. (3) In-depth and multi-dimensional research and analysis of the operational efficiency and influencing factors of high-end equipment manufacturing enterprises, providing ideas and methods for the transformation and upgrading of the industry.

## 2. Literature Review

The existing literature showcases myriad research endeavors, seeking to advance the competitiveness of the high-end equipment manufacturing industry and its enterprises despite a gamut of challenges. Scholars have put forth several propositions, which encompass drawing on the experience of influential nations [11], augmenting the scale of enterprises [12], adept management of business model innovation [13], ameliorating asset management [14], undertaking digital transformation [15,16] and pursuing independent innovation [17], mastering essential technology resources, and fostering talent development [18,19]. Moreover, the "Belt and Road" strategy's historical opportunities can serve as a catalyst, propelling China's high-end equipment manufacturing industry towards "globalization" [20]. Additionally, research has been undertaken on the efficiency of high-end equipment manufacturers in the following areas. (1) Financing efficiency. As per the extant literature, the mean financing efficiency of Chinese equipment manufacturing enterprises is slightly superior to that of conventional manufacturing enterprises [21]. However, it lags behind other strategic emerging industries [22], and the potential of technological advancements in bolstering financing efficiency remains largely untapped. (2) Innovation efficiency. The general innovation efficiency of China's high-end equipment enterprises is not low, with a gradual trend of optimization. However, the innovation efficiency in the technology development phase is relatively low, while the innovation achievement transformation stage exhibits a high level of efficiency. Basic research assumes a crucial role in the progression of China's high-end equipment enterprises. Additionally, localized technological innovation appears more beneficial towards regional development and demands reliance on local capabilities to accomplish technological leapfrogging [13,23]. (3) Operating efficiency. Studies have centered on utilizing financial or non-financial indicators to signify business performance [24]. However, financial performance prioritizes the accomplishments and performance of the enterprise or organization as a whole, whereas operational efficiency accentuates the efficiency level in resource utilization by the enterprise or organization.

Operating efficiency denotes an enterprise's ability to produce a product or offer a service in a more effective manner by optimizing the utilization of its resources. The evaluation of operational efficiency methods has evolved over time and encompasses a range of indicators such as a single efficiency indicator stage (e.g., production efficiency, sales efficiency), a comprehensive efficiency indicator stage (e.g., total factor productivity, economic value added), a non-financial indicator stage (e.g., customer satisfaction, employee satisfaction) [25], and a modeling stage (e.g., data envelopment analysis, stochastic frontier models) [26]. These stages demonstrate that evaluating operational efficiency can aid enterprises in developing suitable evaluation systems to achieve their ultimate corporate objectives and values as the business environment evolves and the evaluation system advances. Financing efficiency, innovation efficiency, and operating efficiency all focus on the efficiency of the enterprise, albeit from different angles. Financing efficiency and innovation efficiency directly impact the enhancement of operating efficiency. Through financing activities, enterprises can acquire funds and increase investment in research and development (R&D) and production to enhance innovation efficiency and operating efficiency. Moreover, improving innovation efficiency can also promote the advancement of production and management efficiency in enterprises, leading to enhanced operating efficiency, which measures overall development and sustainable growth of the enterprise.

The study of factors affecting operational efficiency of organizations has yielded many findings. For instance, the geographical location of airports affects their operational efficiency [27]. Operational strategies of power systems also impact the operational efficiency of wind-hydro power plants [28]. E-commerce technology indirectly improves the operational efficiency of Chinese apparel companies [29], while factors such as firm size, profitability, and higher education affect the dynamic operational efficiency of real estate companies [30]. Additionally, green mergers and acquisitions have improved the operational efficiency of most firms [31]. The positive effects of network size and network centrality on the operational efficiency of technology business incubators (TBIs) in China have been well-documented [32]. Furthermore, government subsidies have been found to improve the operational efficiency of industrial firms by mitigating financing constraints and attracting local investments [33]. Additionally, except for government subsidies, the level of regional internet infrastructure has also been shown to enhance the operational efficiency of cross-border e-commerce firms [34]. The determinants of operational efficiency are diverse, contingent upon the unique characteristics of each organization and the various metrics employed to assess operational efficiency.

The issues in the study of the operating efficiency of high-end equipment manufacturing enterprises are multifaceted. Firstly, the research in this area is limited and lacks depth, with a reliance on financial and non-financial indicators for performance measurement. Secondly, the use of operational efficiency models within this industry has received scant attention in the literature. Finally, the literature exploring the impact of various internal and external factors on the operational efficiency of high-end equipment manufacturing enterprises is yet to be fully developed.

## 3. Materials and Methods

### 3.1. Study Sample Selection and Classification

According to the industry division standard of the SEC and the classification of the high-end equipment manufacturing industry in the "Strategic Emerging Industry Classification Standard (2018)" published by the National Bureau of Statistics [35], A-share listed companies in China were screened for companies whose primary business is rail transportation equipment, marine engineering equipment, aviation equipment, satellite and application equipment, and intelligent manufacturing equipment, excluding ST and ST* companies, as well as those with abnormal data; 103 listed companies were ultimately screened for inclusion in the study sample.

For the study sample by industry, 24 rail transportation equipment enterprises, 17 marine engineering equipment enterprises, 34 aviation equipment enterprises, 20 satellite and

application equipment enterprises, and 8 intelligent manufacturing equipment enterprises were included. By geographical classification [36], 75 enterprises are in the eastern region and 28 enterprises are in the central and western regions.

### *3.2. Variable Selection*

### 3.2.1. Selection of Input–Output Indicators for Operating Efficiency Evaluation

Drawing on the extant literature and taking into account the distinctive features and challenges of the high-end equipment manufacturing industry, as well as the imperative of sustainable enterprise development, this study devised an index system to assess the operational efficiency of such enterprises. To this end, we selected input indicators pertaining to capital, assets, human resources, and R&D. Specifically, we considered operating costs, net value of fixed assets, payroll payable to employees, and R&D investment. As for output indicators, we focused on capital and technology and selected operating income and the number of independent and joint patent applications filed in the current year, as shown in Table 1.

**Table 1.** Selection of input-output indicators for operating efficiency evaluation.

| Type | Indicator | Indicator Explanation |
|---|---|---|
| Input indicators | operating cost (X1) | Capital investment: reflecting the capital invested in the production and operation of the enterprise. |
| | net fixed assets (X2) | Asset inputs: The provision of basic production materials constitutes a fundamental requirement for enterprise development. In particular, high-end equipment manufacturing enterprises rely heavily on specialized equipment, which commands a significant market value. |
| | payroll payable to employees (X3) | Labor input: employee remuneration serves as quantifiable indicators of a company's investment in its labor force. Furthermore, the cultivation of research and development and management personnel constitutes a vital driver of high-end equipment manufacturing development. |
| | R&D input amount (X4) | Technology investment: The high-end equipment manufacturing industry is characterized by advanced technology and high value-added features. In this context, investment in research and development to foster new technological products and innovative solutions represents a crucial source of future efficiency enhancement for enterprises. |
| Output indicators | operating income (Y1) | Monetary output: The monetary revenue generated by an enterprise through the sale of goods or provision of services during a given period represents a key indicator of the enterprise's profitability and sustainability. |
| | the number of patent applications (Y2) | Technical output: The expression of a company's innovation capacity and technological proficiency to a certain extent embodies the high-end equipment manufacturing industry's aptitude for inventiveness and its consciousness in seeking technological or aesthetic exclusivity. |

### 3.2.2. Influencing Factor Variables

- Core explanatory variables: Government subsidies

In this subsection, this paper presents a statistical analysis of the government subsidies received by the sample companies during the period from 2016 to 2021. Over the period spanning 2016 to 2021, the total value of government subsidies awarded to the sample businesses is USD 1.684 billion, USD 1.591 billion, USD 1.75 billion, USD 1.788 billion, USD 2.24 billion, and USD 2.082 billion, respectively, with the subsidy amount for 2020 being particularly high due to the COVID-19 pandemic. Notably, the average value of government subsidies allocated to enterprises located in the eastern region is considerably higher than that of their central and western counterparts (Figure 1). Additionally, as of 2018, the average value of corporate government subsidies in the rail transportation equipment industry surpasses that of other sectors, whereas the average value of such subsidies in the aviation equipment manufacturing industry is markedly lower (Figure 2).

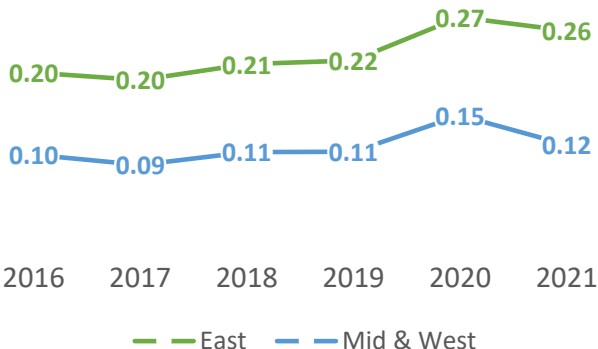

**Figure 1.** Average value of government subsidies for enterprises by regional statistics.

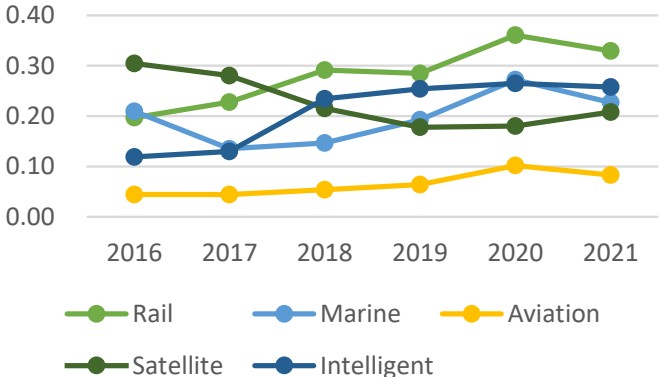

**Figure 2.** Average value of government subsidies for enterprises by industry statistics.

- Control variables

Acknowledging the possibility of other factors influencing the operating efficiency of high-end equipment manufacturing companies, this paper includes the following control variables at the company level and external environment level.

1.  Firm age. Longer-established companies tend to accumulate more capital and expertise in resource allocation, which can positively impact their operational efficiency [37]. However, as companies grow in size and age, they may also face institutional reform and technical innovation challenges that can hinder their efficiency.
2.  Enterprise equity concentration. A higher concentration of equity provides firms with advantages when making critical decisions and also gauges the firm's stability, with a significant positive linear link to the firm's business performance [38].
3.  Regional economic development level. Various regional governments implement different policies to support the growth of their local high-end equipment manufacturing industry. In addition, geographical location and transportation conditions can also have an impact on the efficiency of businesses. Furthermore, the high-end equipment manufacturing industry can act as a catalyst for industrial transformation and upgrading and has a positive impact on regional economic development [39].
4.  Regional openness level. The high-end equipment manufacturing industry highly internationalized, and its growth is closely tied to international market demand and competition. Therefore, a higher degree of regional openness can expand the market size of the local high-end equipment manufacturing industry and encourage international cooperation and exchange, which can contribute to the development of the industry by introducing more foreign technology and capital [40]. However, a higher level of regional openness may also intensify competitive pressure on the local high-end equipment manufacturing industry, requiring local businesses to continuously improve their technical level and product quality to remain competitive.

5.  The regional level of technological development. Higher regional R&D investment funding indicates that local businesses have greater resources and capabilities for technological innovation, which can lead to the faster development and launch of new products and technologies, increasing their market competitiveness. This also indicates an improvement in the region's technology and innovation capabilities, which can foster an innovative culture and atmosphere within the local business community [41]. Furthermore, increased in R&D investment can encourage collaboration between industry, academia, and research, achieving synergistic development of the industrial chain, creating an industrial ecosystem, and further enhancing the competitiveness and influence of the high-end equipment manufacturing industry throughout the entire region.
6.  Regional foreign investment amount. Foreign investment is pivotal in advancing technological advancements in China's manufacturing industry. Huang and Zhang's research shows that OFDI has increased the productivity of Chinese manufacturing firms [42].

Table 2 presents a summary of the explanatory variables, core explanatory variables, and control variables used in this study. It includes the names, symbols, and descriptions of each variable.

**Table 2.** Variable name and definition.

|  | Variable Name | Variable Symbols | Variable Description |
|---|---|---|---|
| Explained variables | Operating efficiency | Efficiency | Operating efficiency values measured by the super-efficient SBM model |
| Core explanatory variables | Government subsidies | GOV | Various forms of government subsidies received by enterprises |
| Control variables | Firm age | AGE | Time of establishment of the enterprise + 1 |
|  | Enterprise equity concentration | CON | Number of shares held by top three shareholders/total number of shares |
|  | Regional economic development level | GDP | Ln (regional GDP) |
|  | Regional openness level | RTV | Total regional import/export/regional economic output |
|  | Regional level of technological development | R&D | Ln (regional R&D investment funds for industrial enterprises above the scale) |
|  | Regional foreign investment amount | INVEST | Ln (total investment of foreign enterprises in the region) |

### 3.3. Data Source

The data of input–output indicators of the explanatory variables of operating efficiency are mainly obtained from the annual statements of each enterprise and the CSMAR database; the data of government support level, age of enterprises, and equity concentration are obtained from the CSMAR database. The data related to regional GDP, regional openness level, regional technology development level, and regional foreign investment amount are obtained from the National Bureau of Statistics and the China Statistical Yearbook.

### 3.4. Model Setting

#### 3.4.1. Super-Efficient SBM Model

The traditional DEA model is a non-parametric evaluation method that does not require any assumptions and pre-set parameter values [43]. This model calculates efficiency values between 0 and 1, and all DEA-effective decision units have an efficiency value of 1. The super-efficient SBM-DEA model estimates efficiency based on a non-radial approach [44], which solves the problems of efficient decision unit ranking and radial

models that do not contain slack variables when measuring inefficient decision units to achieve a more efficient and accurate measurement of decision units.

Assume that there are k decision units, and each decision unit has m inputs and n outputs. If the *p*-th input of the *i*-th decision unit is denoted as $x_{pi}$ ($p = 1, 2, \ldots, m$) and the *q*-th output is denoted as $y_{qi}$ ($q = 1, 2, \ldots, n$), then the super-efficient SBM-DEA model with unguided and variable returns to scale is as follows, where $\rho$ denotes the efficiency value, and when $\rho \geq 1$, it means that the evaluated DMU is effective; $\lambda$ denotes the weight vector; $s_p^-$, $s_q^+$ denote the input and output slack variables, respectively.

$$\min \rho_{SE} = \frac{1 + \frac{1}{m}\sum_{p=1}^{m} s_p^-/x_{pi}}{1 - \frac{1}{s}\sum_{q=1}^{s} s_q^+/y_{qi}}$$

$$s.t. \begin{cases} \sum\limits_{j=1, j\neq i}^{k} x_{pj}\lambda_j - s_p^- \leq x_{pi} \\ \sum\limits_{j=1, j\neq i}^{k} y_{qj}\lambda_j + s_q^+ \geq y_{qi} \\ \sum\limits_{j=1, j\neq i}^{k} \lambda_j = 1 \end{cases} \tag{1}$$

$$\lambda, s^-, s^+ \geq 0$$

$$p = 1, 2, \ldots, m; q = 1, 2, \ldots, n; j = 1, 2, \ldots k (j \neq i)$$

### 3.4.2. Tobit Regression Model

Given that the explanatory variables utilized in this paper are predicated on the operating efficiency of publicly traded firms in the high-end equipment manufacturing industry as gauged through the super-efficient SBM model, and given that the operating efficiency values are truncated data, an OLS model used for empirical verification would give rise to biases and inconsistencies in the ultimate findings, rendering them incapable of providing an accurate portrayal of the data's veracity. To overcome the issue of limited dependent variable, the Tobit model can employ maximum likelihood estimation to guarantee the reliability of the results.

$$Y_i = \begin{cases} \beta^t X_i + \mu_i, & \beta^t X_i + \mu_i > 0 \\ 0, & \beta^t X_i + \mu_i \leq 0 \end{cases} \tag{2}$$

where $Y_i$ is the explanatory variable, and if the value of $Y_i$ is less than or equal to 0, then $Y_i = 0$, i.e., the left-hand side is truncated; $\beta^t$ is the parameter vector, $X_i$ is the explanatory variable vector, and $\mu_i$ obeys normal distribution. The following regression equation is constructed to investigate the effects of government subsidies as well as other control variables on the operating efficiency of high-end equipment manufacturing firms.

$$Efficiency_{it} = \alpha_0 + \beta_1 GOV_{it} + \beta_2 AGE_{it} + \beta_3 CON_{it} + \beta_4 GDP_{it} + \beta_5 RTV_{it} + \beta_6 R\&D_{it} + \beta_7 INVEST_{it} + \sum Year + \sum Industry + \varepsilon_{it} \tag{3}$$

where $Efficiency_{it}$ is the operating efficiency value of the ith listed company in period $t$, $\alpha_0$ is the constant term, $\beta_i$ is the coefficient to be estimated, *GOV* represents the level of government support, *AGE* represents the year of company establishment, *CON* represents the concentration of company equity, *GDP* represents the gross domestic product of the province where the company is located, *RTV* represents the level of regional openness, *R&D* represents the regional level of technological development, *INVEST* represents the enterprise regional foreign investment, *Year* is the year fixed effect, *Industry* is the industry fixed effect, and $\varepsilon_{it}$ is the random disturbance term.

## 4. Results and Discussion

### *4.1. Operating Efficiency Measurement*

4.1.1. Overall Operating Efficiency Analysis

Based on the Table 1 indicator system, using Equation (1), setting variable scale efficiency and unguided approach, the DEA-Solver software was applied to measure the operating efficiency of the selected study sample separately by year for the period 2016–2021. This software is celebrated for its user-friendly application and expeditious computational velocity.

The average operating efficiency of the entire enterprise sample during the years 2016–2021 are as follows: 0.6418, 0.6765, 0.7223, 0.6670, 0.6989, and 0.7762, respectively. It is apparent that that the overall operating efficiency of high-end equipment manufacturing enterprises exhibits a state of non-DEA validity. The lower operating efficiency signifies the presence of specific issues in the current business operations of high-end equipment manufacturing enterprises, such as inadequate input–output redundancy. The average value of the efficiency of the entire sample from 2016 to 2018 demonstrates a continuous increase. During this period, high-end equipment manufacturing enterprises exhibited an overall improvement in operational efficiency, leading to promising prospects for the flourishing development of the high-end equipment manufacturing industry. The auspicious growth prospects of the high-end equipment manufacturing industry are supported by the supply-side structural reform policy of "three go, one drop, one supplement", which targets the reduction of excess industrial capacity, deleveraging in the corporate sector, destocking of property inventories, cost reduction for businesses, and remediation of weak links in the economy [45]. This policy has conferred momentum upon the industry's expansion. Under the influence of multiple unfavorable factors, such as the severe and complex international environment and the increased downward pressure on the domestic economy during 2019–2021, the operating efficiency of some enterprises experienced significant volatility. However, since the calculated efficiency is relative efficiency, the overall operating efficiency has actually increased after a period of decline.

In Table 3, the calculated efficiency values have been partitioned, and it is observed that, apart from the partition with efficiency values between 0.80 and 1, the number of companies is relatively evenly distributed across the other partitions. The number of highly efficient companies that are close to the DEA efficiency threshold of [0.8,1) is relatively small, with only one company in some years falling within this range, which has created a bottleneck for the growth of efficiency. The proportion of enterprises with efficiency values below 0.6 ranges from 41.75 to 62.14 percent, indicating the majority of businesses operate inefficiently and are in a non-DEA effective low-efficiency stage. In terms of the overall development trend, the proportion of companies reaching DEA effectiveness increased, from 32% in 2016 to 40% in 2021. Additionally, the proportion of companies in the high and medium efficiency intervals also increased, from a total proportion of 9.7% to 18.7%. Meanwhile, the proportion of companies in the low-efficiency interval has decreased, indicating that some companies are gradually reaching higher efficiency levels.

4.1.2. Operating Efficiency Statistics by Industry

Figures 3 and 4 depict the average value of operating efficiency and the proportion of effective DMUs in different industries and years, respectively. The railway transportation equipment manufacturing industry has maintained a relatively stable efficiency level, which is similar to the trend of the marine engineering equipment industry but with a higher average efficiency. The efficiency mean of the marine engineering industry is the lowest among the five industries, except for the year 2021, and the he proportion of effective DMUs is also the smallest except for the year 2020. The aviation equipment industry has the highest proportion among the total sample, with efficiency mean only higher than the marine engineering equipment manufacturing industry in the early years, and in 2021, it has the lowest efficiency mean among the five industries. There are 20 satellite and application enterprises, and their efficiency mean from 2016 to 2019 is only second to the

intelligent manufacturing equipment manufacturing industry, and in 2020, the efficiency mean is the lowest among the five industries. The proportion of intelligent manufacturing companies is the smallest, yet their efficiency values were greater than 1 from 2017 to 2021, and the proportion of effective DMUs is also highest, reaching 50% and 75%. This indicates that at least half of the companies in this industry are in DEA-effective state. The reason is that intelligent manufacturing equipment enterprises are developing rapidly, have sufficient R&D investment, obtain multiple patented technologies, and have a wide application of product technologies.

**Table 3.** Operating Efficiency Segment Statistics.

| Efficiency Interval | | 2016 | | 2017 | | 2018 | | 2019 | | 2020 | | 2021 | |
|---|---|---|---|---|---|---|---|---|---|---|---|---|---|---|
| | | Q | P | Q | P | Q | P | Q | P | Q | P | Q | P |
| Low efficiency | [0,0.2) | 20 | 19.42% | 20 | 19.42% | 8 | 7.77% | 9 | 8.74% | 12 | 11.65% | 10 | 9.71% |
| | [0.2,0.4) | 20 | 19.42% | 26 | 25.24% | 24 | 23.30% | 27 | 26.21% | 29 | 28.16% | 15 | 14.56% |
| | [0.4,0.6) | 20 | 19.42% | 9 | 8.74% | 19 | 18.45% | 28 | 27.18% | 16 | 15.53% | 18 | 17.48% |
| Medium efficiency | [0.6,0.8) | 9 | 8.74% | 7 | 6.8% | 15 | 14.56% | 9 | 8.74% | 9 | 8.74% | 12 | 11.65% |
| High efficiency | [0.8,1) | 1 | 0.97% | 6 | 5.83% | 2 | 1.94% | 1 | 0.97% | 1 | 0.97% | 7 | 6.80% |
| Super efficiency | [1,4) | 33 | 32.04% | 35 | 33.98% | 35 | 33.98% | 29 | 28.16% | 36 | 34.95% | 41 | 39.81% |

Note: Q = Quantity; P = Percentage.

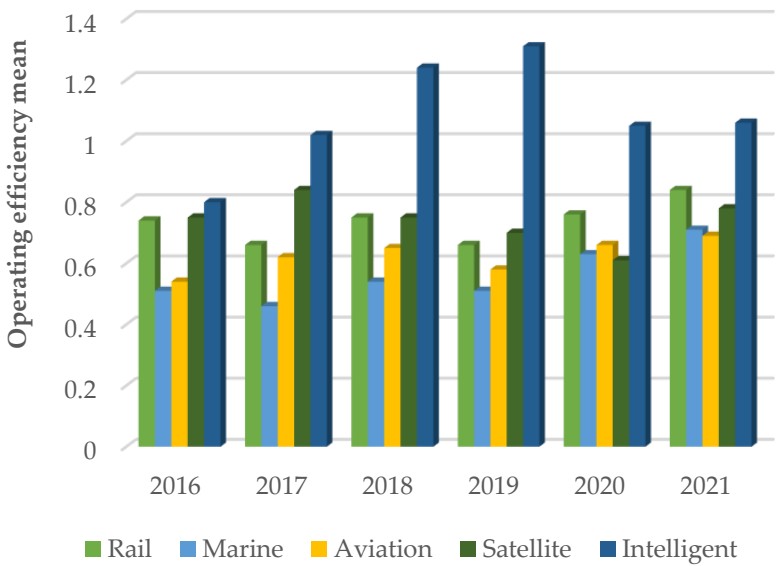

**Figure 3.** Average operating efficiency by industry statistics.

4.1.3. Operating Efficiency Statistics by Region

Except for a minor decline in 2019, the average operating efficiency in both the East and Midwest regions has generally been increasing. From 2016 to 2019, the average value of enterprise operating efficiency in the eastern region was higher than that in the central and western regions, but from 2020 to 2021, the central and western regions surpassed the eastern region. Additionally, the effective DMU ratio for the mid- and west region is gradually increasing, indicating that businesses in these regions are growing rapidly and have great development potential (Figure 5).

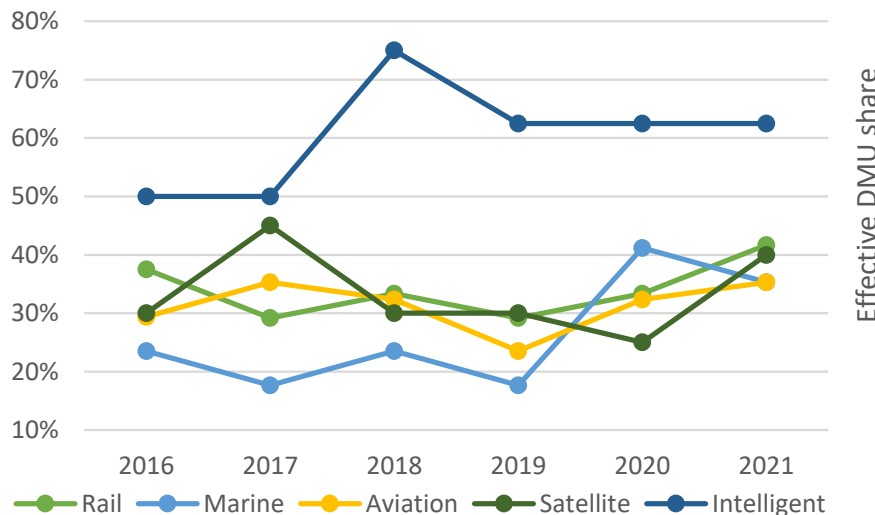

**Figure 4.** Percentage of effective DMUs by industry statistics.

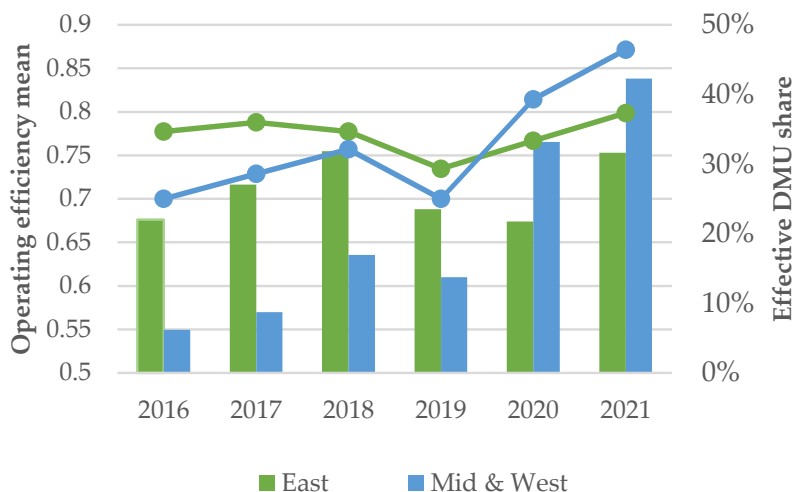

**Figure 5.** Average efficiency and effective DMUs share by region.

4.1.4. Analysis of Slack Variables

In this subsection, analyze the redundancy variables for different DMUs, where X1, X2, X3, X4, Y1, and Y2 represent the input-output variables: operating cost, net fixed assets, employee compensation payable, R&D investment amount, operating income, and the number of patents applied in the year. s− and s+ represent the proportion of reduction and increase needed to reach the target value, respectively.

Under the cross-section of 2021, The top 8 firms for which DEA was consistently effective during the study period and the 8 firms for which efficiency values were consistently below 0.4 during the study period were taken for the analysis of under-redundant variables (there are 15 DMUs for which DEA is consistently valid and 8 DMUs for which efficiency values are consistently below 0.4 for each year).

Table 4 shows that even for the DEAs that are effective according to DEA, there are still variables that exhibit redundancy and deficiency. The top-ranked effective DMU (300173) needs to reduce employee compensation payable (X3) by 10.59% and increase R&D investment (X4) by 8.39% to reach the optimum level. In addition, four enterprises have redundancy in the variable of employee compensation payable (X3), indicating the existence of staff redundancy, which requires enterprises to avoid waste and redundancy of resources by improving management system, adjusting the organizational structure of employees, the compensation structure of enterprises, or improving employee efficiency through training.

Two enterprises (600850, 000063) have redundancy in both R&D investment (X4) and the number of patents applied for (Y2). R&D innovation is crucial for sustainable development and requires enterprises to improve their R&D strategies, increase the efficiency of R&D, and enhance the utilization of R&D funds.

All eight DMUs that were found to be valid exhibit various degrees of deficiency in the number of patents applied for (Y2) variable. This indicates these companies possess insufficient R&D output and low operating efficiency when compared to other companies. If these enterprises aspire to improve their operating efficiency, they must focus on enhancing their R&D quality, including R&D expenditure and personnel. By strengthening R&D investment and R&D quality, optimizing operating management, these enterprises can improve their overall operation level and operating efficiency. Additionally, three enterprises (600592, 000880, 002480) possess redundancy in four input variables simultaneously and therefore need to enhance their organizational structure, augment their resource allocation capability, and improve their overall efficiency.

**Table 4.** Statistics on under-redundant variables (unit: %).

| | | | Under-Redundant Variables for Effective DMUs | | | | | | | | | Under-Redundant Variables for Non-Effective DMUs | | | | | |
|---|---|---|---|---|---|---|---|---|---|---|---|---|---|---|---|---|---|
| R | DMU | RA | Y1 | Y2 | X1 | X2 | X3 | X4 | R | DMU | RA | Y1 | Y2 | X1 | X2 | X3 | X4 |
| 1 | 300173 | s− | | | | | −10.59 | | −1 | 601890 | s− | | | | −20.86 | −44.58 | |
| | | s+ | | | | | | 8.39 | | | s+ | | 999.9 | | | | |
| 2 | 601766 | s− | | | −66.05 | −56.56 | −77.04 | −73 | −2 | 002520 | s− | | | | −82.31 | −48.34 | |
| | | s+ | 69.69 | | | | | | | | s+ | | 999.9 | | | | |
| 3 | 600850 | s− | | −999.9 | −94.34 | | −84.7 | −62.72 | −3 | 600592 | s− | | | −15.21 | −5.98 | −35.05 | −15.53 |
| | | s+ | 93.38 | | | 79.94 | | | | | s+ | | 927.76 | | | | |
| 4 | 000063 | s− | | −140.38 | | | −74.9 | −81.49 | −4 | 300424 | s− | | | | | −42.12 | |
| | | s+ | 24.3 | | | 10.01 | | | | | s+ | | 778.33 | | | | |
| 5 | 600973 | s− | | −58.55 | −24.75 | −23.7 | | | −5 | 600151 | s− | | | | −23.1 | −63.11 | −12.02 |
| | | s+ | 21.55 | | | | 5.58 | | | | s+ | | 381.79 | | | | 0 |
| 6 | 300095 | s− | | −118.73 | | | | | −6 | 002369 | s− | | | −6.06 | −32.65 | −65.9 | |
| | | s+ | | | 39.88 | 9.44 | 150.06 | 4.02 | | | s+ | | 284 | | | | |
| 7 | 000039 | s− | | | −53.58 | −67.31 | −62.07 | | −7 | 000880 | s− | | | −17 | −19.42 | −71.45 | −45.76 |
| | | s+ | 56.78 | | | | | | | | s+ | | 184.48 | | | | |
| 8 | 600406 | s− | 0 | 0 | 0 | 0 | 0 | −24.92 | −8 | 002480 | s− | 0 | 0 | −27.37 | −49.58 | −55.47 | −52.37 |
| | | s+ | 0 | 0 | 15.46 | 20.25 | 0 | 0 | | | s+ | 0 | 69.71% | 0 | 0 | 0 | 0 |

### 4.2. Analysis of Factors Influencing Operating Efficiency

#### 4.2.1. Variable Descriptive Statistics

Table 5 reveals that the lowest level of government support is only CNY 2000, while the highest is CNY 3.498 billion, with a standard deviation of 3.38, indicating a relatively stable level of government support and a moderate level of data dispersion. However, there is a phenomenon of bipolar differences between individual companies, suggesting that there are significant variations in the level of government support for different fields and enterprises, which may be related to factors such as industry, enterprise size, and region. The selected companies have an average age of 19.7 years with a standard deviation of 5.24 years, indicating a relatively small in the years of establishment of high-end equipment manufacturing companies in China. The equity distribution of the sample companies is highly uneven, with significant variability in equity concentration. Data for regional GDP, regional opening-up level, regional R&D investment, and regional foreign investment exhibit substantial differences due to geographical conditions and resource distribution, resulting in significant regional disparities.

#### 4.2.2. Person Correlation Analysis

From Table 6, it can be seen that except for a few variables with relatively large coefficients, the absolute values of the correlation coefficients between other variables were less than 0.3. This indicates that there is no obvious collinearity among the variables in the model, and Tobit regression could be performed.

**Table 5.** Descriptive statistics of the raw data of the variables.

| Variable | Unit | Sample Size | Average | SD | Min | Max |
|---|---|---|---|---|---|---|
| GOV | Billion CNY | 618 | 1.02262 | 3.3803 | 0.00002 | 34.98 |
| AGE | Year | 618 | 19.70 | 5.24 | 9 | 36 |
| CON | % | 618 | 43.57 | 14.21 | 6.10 | 83 |
| GDP | Billion CNY | 618 | 58,089.35 | 32,813.87 | 11,477.2 | 124,719.5 |
| RTV | % | 618 | 0.4801 | 0.2775 | 0.0270 | 0.9765 |
| R&D | Billion CNY | 618 | 1068.507 | 872.6531 | 55.6853 | 2902.185 |
| INVEST | Billion CNY | 618 | 52.20388 | 29.23975 | 1 | 101 |

Note: CNY is yuan, the Chinese currency.

**Table 6.** Correlation analysis of explanatory variables.

| | GOV | AGE | CON | GDP | RTV | R&D | INVEST |
|---|---|---|---|---|---|---|---|
| GOV | 1.0000 | | | | | | |
| AGE | 0.0459 | 1.0000 | | | | | |
| CON | 0.0305 | 0.0504 | 1.0000 | | | | |
| GDP | 0.0436 | 0.0294 | −0.1396 | 1.0000 | | | |
| RTV | 0.0349 | −0.2359 | −0.0837 | 0.2471 | 1.0000 | | |
| R&D | 0.0479 | 0.0652 | −0.1584 | 0.2440 | 0.3172 | 1.0000 | |
| INVEST | −0.0093 | 0.1895 | −0.0594 | −0.2526 | −0.5618 | −0.2179 | 1.0000 |

### 4.2.3. Baseline Regression Results

Using Equation (3), Stata software conducted Tobit regressions. Column 1 of Table 7 demonstrates the results of the full-sample regression without control variables, and the regression coefficient of GOV is statistically significant at the 1% level when controlling for year-fixed effects and industry-fixed effects. After all control variables have been included, column 2 demonstrates that the regression coefficient of GOV is positively significant at the 1% level. With the exception of the regional foreign investment variable, all other control variables are significantly correlated with operating efficiency. Robustness tests were conducted in column 3 and 4, where the main explanatory variables for the regression analysis were replaced with the ratio of government subsidies to total assets (GOVA) and the natural logarithm of the number of government subsidies (LNGOV), respectively. At the 1% level of significance, the regression coefficients of GOVA and LNCOV are both positively significant. The results suggest that the conclusion that government subsidies promote business operating efficiency is robust to the way in which it is measured.

1. The level of government support is positively correlated with operating efficiency and significant at the 1% level of confidence. This suggests that the provision of government support, such as indirect tax subsidies and incentive funds, can aid firms in mitigating the expenses and uncertainties involved in research and development, thereby spurring technological innovation and product development. Nevertheless, it is imperative that the level of government assistance is measured and moderate. Overbearing government intervention may impede the market's self-governing regulatory mechanism, impairing firms' autonomous innovation and operational prowess.

2. At the 1% level of confidence, both the age of the business and the concentration of equity statistically significant. Given the nature of high-end equipment, which often involves high technology and high-value products, enterprises must possess advanced technological innovation and management capabilities as well as a deep reservoir of R&D experience and knowledge. Furthermore, the employees of the business must be capable of comprehending market demand and customer needs, skills that usually require a long period of accumulation and practice. In enterprises with a higher concentration of equity, minority shareholders typically wield more influence and decision-making power, which enables businesses to make decisions more swiftly and respond more quickly to market changes. Additionally, companies with a higher concentration of ownership generally possess more stable and long-term

growth strategies than those with fragmented ownership, allowing them to better plan and manage their resources, and ultimately improve their operating efficiency.

3. The regional GDP is significantly and positively correlated at the 5% level of confidence. The GDP of a region reflects the overall level of economic development, and governments in high GDP regions typically invest more resources in supporting the local businesses. This may include better infrastructure, more policy support and a better talent pool, all of which contribute to improving the operating efficiency of businesses. Additionally, a higher GDP implies a greater demand in the market, which may help increase the sales volume and revenue of the businesses.

4. The degree of regional openness is significantly and positively correlated with the operating efficiency of high-end equipment manufacturing enterprises, as confirmed at a confidence level of 1%. The relationship reflects the level of communication and trade between a region and the outside world. Given the necessity of high-end equipment manufacturers to compete in the global market, an open market environment can bring about more competition, opportunities, and potential partners, all of which can promote the development and growth of high-end equipment manufacturing enterprises. Moreover, high-end equipment manufacturing also requires numerous critical raw materials and technologies, which are typically imported from overseas. Thus, the higher the degree of openness in a region, the easier it is for these enterprises to acquire essential resources and technologies, thereby facilitating their development and growth.

5. The regional level of technological development has been found to be significant at the 1% confidence level, but with a negative correlation. If an area increases its R&D expenditure, it may encourage high-end equipment manufacturing enterprises to increase their efforts in technological research and development. However, due to the relatively long production cycle of high-end equipment manufacturing and the complex components and technologies involved in the production process, a large amount of R&D investment and time is required to obtain new technologies and improve product quality. Investments in innovation and R&D often take a long time to yield returns, so even with more R&D investment, it may not immediately translate into an improvement in operational efficiency. In this process, enterprises need to bear considerable costs and risks. Therefore, excessive regional R&D investment may lead to a decrease in overall operational efficiency of the enterprise.

4.2.4. Heterogeneity Analysis

In Table 8, columns 1 through 5 of the table contain the findings from a regression analysis of the five industry groups, which are, respectively, the rail transportation equipment industry, the industry for marine engineering the aviation equipment industry, and the satellite and application equipment industry and intelligent manufacturing equipment. The intelligent manufacturing industry, which incurs high costs for technological innovation and R&D investment, did not exhibit any significant findings. While the average government subsidies for businesses in this industry are second only to those in the rail transportation industry, the number of subsidies may not adequately meet the businesses' actual needs, failing to serve as an effective promotional tool. Moreover, the government subsidies may lead to unfair competition among businesses, thereby impeding market efficiency. These factors may impact the relationship between government subsidies and operating effectiveness, resulting in a negligible effect of government subsidies on the sector. The remaining four industries demonstrated significant findings at different confidence levels, namely 1%, 5%, 5%, and 10%.

Columns 6 and 7 in Table 8 pertain to the grouped regressions for the eastern region and the central and western regions, respectively. The enterprises in the eastern region demonstrated significant findings at a 1% confidence level, whereas those in the central and western regions did not exhibit any significant findings. These are two potential reasons for this. Firstly, the technology level in the manufacturing sector of the central and western

regions generally lags behind that of the eastern region, and the average operating efficiency and government subsidies of enterprises in the eastern region are higher than those in the central and western regions. The lower government subsidies may not sufficiently offset the effects of technological shortcomings, which could impede the improvement of operating efficiency. Secondly, the manufacturing market in the central and western regions is relatively small, less competitive, and the allocation of resources such as talent, capital, and technology is inadequate compared to the eastern regions. These factors may limit the development of businesses and the effective utilization of government subsidies.

**Table 7.** Baseline model regression results.

| Variable | 1 | 2 | 3 | 4 |
|---|---|---|---|---|
| GOV | 0.0438 *** (8.44) | 0.0361 *** (6.64) | | |
| AGE | | 0.0109 *** (2.74) | 0.0133 *** (3.14) | 0.0099 ** (2.46) |
| CON | | 0.0039 *** (2.71) | 0.0073 *** (5.29) | 0.0049 *** (3.43) |
| GDP | | 0.2390 ** (2.30) | 0.2400 ** (2.25) | 0.2315 ** (2.19) |
| RTV | | 0.2857 *** (3.10) | 0.3835 *** (4.11) | 0.3013 *** (3.20) |
| R&D | | −0.2018 *** (−3.02) | −0.2084 *** (−3.04) | −0.1930 *** (−2.84) |
| INVEST | | −0.0009 (−1.09) | −0.0011 (−1.30) | −0.0009 (−1.08) |
| GOVA | | | 10.6772 *** (3.23) | |
| LNGOV | | | | 0.0706 *** (4.76) |
| Year | Y | Y | Y | Y |
| Industry | Y | Y | Y | Y |
| N | 618 | 618 | 618 | 618 |

Note: (1) Inside the parentheses is the t-statistic; (2) **, *** indicate significant at 5% and 1% confidence level, respectively.

**Table 8.** Regression results by industry and by region.

| | 1 | 2 | 3 | 4 | 5 | 6 | 7 |
|---|---|---|---|---|---|---|---|
| GOV | 0.0159 * (1.87) | 0.0504 ** (2.52) | 0.1381 ** (2.37) | 0.0430 *** (5.35) | −0.0428 (−0.99) | 0.0364 *** (6.23) | 0.1279 (1.44) |
| Control | Y | Y | Y | Y | Y | Y | Y |
| Year | Y | Y | Y | Y | Y | Y | Y |
| Industry | N | N | N | N | N | Y | Y |
| N | 144 | 102 | 204 | 120 | 48 | 450 | 168 |

Note: *, **, *** indicate significant at 10%, 5% and 1% confidence level respectively.

## 5. Conclusions and Recommendations

The rapid development of the high-end equipment manufacturing industry and the challenges it faces necessitate that China's high-end equipment manufacturing companies enhance their overall competitiveness. This paper presents conclusions drawn from an analysis of the research sample's efficiency evaluation and influence factors. Firstly, the operational efficiency of listed enterprises in China's high-end equipment manufacturing industry is low, with the average efficiency value of the entire sample enterprises around 0.7, placing them in the middle-efficiency stage of the non-DEA effective state. Only about 30% of enterprises are in a DEA effective state, but the effective DEA ratio is increasing annually. However, the number of enterprises in the low-efficiency range has decreased, and the number of enterprises in the medium and high-efficiency range has increased.

Secondly, the average efficiency value in different industries varies significantly, with intelligent manufacturing enterprises demonstrating a more prominent average efficiency value and a greater proportion of effective DEA enterprises compared to other industries. Conversely, the marine engineering equipment manufacturing industry has a low mean value, placing it in the low-efficiency stage and exhibiting the lowest proportion of DEA-effective firms. Thirdly, in the eastern region, the average operating efficiency value remains relatively stable, while in the central and western regions, both the average efficiency value and the proportion of effective DEA enterprises are increasing annually. Lastly, when both inputs and outputs are taken into account, both DEA-effective and ineffective businesses have varying degrees of redundancy deficiencies in different variables, with less efficient businesses demonstrating a severe deficiency in the output of the number of patent applications.

At the government level, the following recommendations are proposed based on the aforementioned findings: (1) To facilitate the expansion of high-end equipment manufacturing businesses and foster scientific research and innovation, the government should devise relevant policies and provide specific subsidies to eligible enterprises in various ways, particularly in the central and western regions. However, the subsidies should be moderate in amount, as only moderate government subsidies can effectively stimulate business innovation. Excessive government subsidies not only dampen enterprise enthusiasm but also result in unfair market competition. (2) The government can support the growth and development of businesses by promoting international trade and enhancing regional openness. Through trade policies and other means, the government can encourage more foreign investment and multinational corporations to invest in and establish factories to support the growth of industrial chains and clusters. The government can also promote exports and increase the market share of local businesses. (3) Due to the relatively low level of technology and enterprise scale, China's high-end equipment manufacturing sector has not yet achieved an overall scale effect or cluster effect. Agglomerational development can promote the development of collaborative innovation in high-end equipment manufacturing [46], local governments should conduct in-depth analyses and develop plans that account for local circumstances to strengthen regional synergistic growth, foster the development of high-end equipment manufacturing enterprise clusters, and create scale effects. Additionally, local governments should encourage the formation of industrial alliances and promote cooperation and exchange between high-end equipment manufacturing enterprises to facilitate resource sharing and complementary advantages and to establish an integrated supply chain. Industrial alliances can also provide technical assistance, market intelligence, and talent development services to assist businesses in enhancing their technical and operational standards.

At the enterprise level, the following recommendations are proposed based on the aforementioned findings: (1) Enhancing technological innovation capacity augmenting the application of technology in products is paramount. Inefficient enterprises suffer from inadequate patent output, and China's high-end equipment manufacturing industry predominantly relies on low-end products. The deficiency of high-end products in domestic markets necessitates their importation. Technical barriers impede the development of aviation engines, high-end chips, and other product technologies that other countries have mastered. These limitations hinder improvement in operating efficiency. Thus, high-end equipment manufacturing companies should fortify their research and development (R&D) capabilities, cultivate independent innovation capacity, and make significant headway in fundamental technology issues. Moreover, optimization of R&D investment and management can enhance the efficiency of the use of R&D funds. By promoting the convergence of technology and industry, docking technical achievements with market demand and augmenting the capacity for applying technology to products, companies can significantly increase their operating efficiency. (2) Alleviating input redundancy and output shortage are critical for both highly efficient and poorly efficient organizations, as indicated by Table 4. Organizational management and resource allocation capabilities should be

strengthened to minimize resource waste and maximize resource utilization by enhancing internal management mechanisms and optimizing processes. Such efforts can achieve the objective of minimal input and maximum output. (3) Scientifically and rationally promote top-level design of equity structure. The concentration and dispersion degree of equity have impacts on various aspects of corporate governance structure, strategic decision-making, and business development [47]. For knowledge-intensive industries such as high-end equipment manufacturing, a higher concentration of equity can improve the decision-making and resource allocation efficiency of the company, and accelerate technological innovation and product development, thus enhancing the overall operating efficiency of the company. However, blindly concentrating equity in a few major shareholders may result in a "one-man show" phenomenon. Therefore, it is necessary to promote the top-level design of equity structure in a reasonable way, by establishing scientific and effective internal governance mechanisms to ensure the scientific and fair decision making of the company. Only in this way can the sustainable development of the company be effectively guaranteed.

**Author Contributions:** Methodology, Y.Z.; writing—review & editing, M.L. All authors have read and agreed to the published version of the manuscript.

**Funding:** This research received no external funding.

**Institutional Review Board Statement:** Not applicable.

**Informed Consent Statement:** Not applicable.

**Data Availability Statement:** The data presented in this study are openly available in CSMAR (gtarsc.com), the National Bureau of Statistics (data.stats.gov.cn) and the China Statistical Yearbook.

**Conflicts of Interest:** The authors declare no conflict of interest.

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
