# Peer review of "Enhancing Operating Efficiency in China’s High-End Equipment Manufacturing Industry: Insights from Listed Enterprises"

_sustainability, doi:10.3390/su15118694_

Round 1

Reviewer 1 Report

I have found some major points which should be incorporated to enhance the quality of the manuscript.

1. One of the most important things is the novelty of the study. What are the significant contributions of the paper? The authors should mention this in the abstract. Which factors make this research different from the others?

2. Separate paragraphs are needed for the following points. a) Novelty of the study, b) Research questions, c) Research Gaps.

3. Literature review section is poorly written. The authors should explain the context of the study under the view of existing literature. It should be explained in such a way that it would match the keywords also. Moreover, no citation of 2022 was provided and only one citation of 2023 is there. 

4. Explain the model properly. The decision variables and parameters are not mentioned in optimization model 1.

5. No solution approach is provided to solve the optimization model.

6. The necessity of the regression model and its relationship with the optimization model was not properly explained.

The paper should be majorly revised according to the above points.

Some minor modifications are required to improve the English.

Author Response

请参考附件。

Reviewer 2 Report

The following comments need to be addressed before publication.

1) Lines 30-43 provide appropriate references.

2) Line 54 - Who is scholars? Rewrite the sentence.

3) Line 115 - Rewrite the sentence "This paper hopes to make the following improvements."

4) According to Table 1, why did the authors choose 04 input indicators?

5) Lines 152-156 provide appropriate references.

6) Why did the authors use DEA-Solver software? 

Extensive editing of the English language required

Author Response

请参阅附件。

Reviewer 3 Report

The authors present a study based on practical guidelines to improve the operational efficiency of equipment manufacturing enterprises. However, there are some flaws in the manuscript that must be resolved so that it can be published in the Sustainability journal.

(1) What is the difference between Figure 1 and Figure 3 since they have the same statement, in the same way, between Figure 2 and Figure 4?

(2) Tables 2, Table 5, and Table 6 are not discussed in the text of the manuscript. Table 8 should be well specified in the text.

(3) On line 469 it says: column 6 and 7, from where?

(4) Improve the discussion of Table 3 in the text as it is very vague.

(5) The paragraph from lines 154 to 156 is confusing in detail.

(6) References must be easily searchable, even if access is not open, for example reference 8, reference 15, and reference 25.

(7) There are punctuation problems in the manuscript.

Round 2

Reviewer 1 Report

The manuscript can be accepted for publication.

The manuscript can be accepted for publication.

Reviewer 2 Report

The authors have incorporated the comments. 

Dear Editor,

Now the paper is suitable for publication.

Reviewer 3 Report

Dear Authors,

I appreciate that you have taken the time to attend to the suggestions to improve your manuscript.

I saw that they also made improvements and additions in the text in the manuscript.

It will only be necessary to make improvements in the wording that the editors will already be indicating.